# Incommensurately Modulated Crystal Structure and Photoluminescence Properties of Eu_2_O_3_- and P_2_O_5_-Doped Ca_2_SiO_4_ Phosphor

**DOI:** 10.3390/ma13010058

**Published:** 2019-12-20

**Authors:** Hiromi Nakano, Shota Ando, Konatsu Kamimoto, Yuya Hiramatsu, Yuichi Michiue, Naoto Hirosaki, Koichiro Fukuda

**Affiliations:** 1Cooperative Research Facility Center, Toyohashi University of Technology, Toyohashi 441-8580, Japan; 2Department of Applied Chemistry and Life Science, Toyohashi University of Technology, Toyohashi 441-8580, Japan; s183404@edu.tut.ac.jp (S.A.); k163413@edu.tut.ac.jp (K.K.); 3Department of Life Science and Applied Chemistry, Nagoya Institute of Technology, Nagoya 466-8555, Japan; 30411131@stn.nitech.ac.jp (Y.H.); fukuda.koichiro@nitech.ac.jp (K.F.); 4National Institute for Materials Science, Tsukuba 305-0044, Japan; michiue.yuichi@nims.go.jp (Y.M.); hirosaki.naoto@nims.go.jp (N.H.)

**Keywords:** optical materials, phase transitions, dicalcium silicate, phase compositions, incommensurate structures

## Abstract

We prepared four types of Eu_2_O_3_- and P_2_O_5_-doped Ca_2_SiO_4_ phosphors with different phase compositions but identical chemical composition, the chemical formula of which was (Ca_1.950_Eu^3+^_0.013_☐_0.037_)(Si_0.940_P_0.060_)O_4_ (☐ denotes vacancies in Ca sites). One of the phosphors was composed exclusively of the incommensurate (IC) phase with superspace group *Pnma*(0*β*0)00*s* and basic unit-cell dimensions of *a* = 0.68004(2) nm, *b* = 0.54481(2) nm, and *c* = 0.93956(3) nm (*Z* = 4). The crystal structure was made up of four types of β-Ca_2_SiO_4_-related layers with an interlayer. The incommensurate modulation with wavelength of 4.110 × *b* was induced by the long-range stacking order of these layers. When increasing the relative amount of the IC-phase with respect to the coexisting β-phase, the red light emission intensity, under excitation at 394 nm, steadily decreased to reach the minimum, at which the specimen was composed exclusively of the IC-phase. The coordination environments of Eu^3+^ ion in the crystal structures of β- and IC-phases might be closely related to the photoluminescence intensities of the phosphors.

## 1. Introduction

Rare-earth doped dicalcium silicate (Ca_2_SiO_4_, C_2_S) polymorphs have been the subject of extensive study because of the promising applicability of stabilized high-temperature modifications to the phosphor materials of white light-emitting LEDs [1,2,3,4,5,6,7,8]. The red light emission is characteristic of the Eu^3+^-activated phosphors, due to the transition of the ^5^D_0_–^7^F_2_ for Eu^3+^ ion [3,4]. The photoluminescence (PL) originates from the 4f–4f dipole transitions and, hence, the emission wavelengths are nearly the same among the various types of phosphors with different host materials. However, the PL intensity has been tunable for Eu^3+^-activated lithium tantalite-based phosphors by controlling the coordination environments of the Eu^3+^ ion [9,10]. Accordingly, we speculated that the PL intensities could be also tunable for the Eu^3+^-activated C_2_S phosphors, depending on the polymorphs stabilized at ambient temperature.

The C_2_S polymorphs established so far are, in order of increasing temperature, γ (orthorhombic), β (monoclinic), α’_L_ (orthorhombic), α’_H_ (orthorhombic), and α (trigonal) [11]. The transformation temperatures during the heating process are 963 K for β-to-α’_L_, 1433 K for α’_L_-to-α’_H_, and 1698 K for α’_H_-to-α. The phase change from α’_L_ to β on cooling has been reported to be thermoelastic martensitic [12]. Thus, the two phases coexisted at temperatures between the transformation starting temperature (=*M*_s_) and finishing temperature (=*M*_f_), both of which steadily decreased, even below ambient temperature (=*T*_a_), with increasing concentration of dopants in C_2_S. The stabilized phases at *T*_a_ systematically changed from β (*T*_a_ < *M*_f_), β + α’_L_ (*M*_f_ < *T*_a_ < *M*_s_) to α’_L_ (*M*_s_ < *T*_a_) with increasing concentrations of foreign ions such as Sr^2+^ or P^5+^ [13,14]. In the crystal structure of β-Ca_2_SiO_4_ there are two types of Ca sites. One (Ca1 site) is surrounded by seven O atoms, and the other (Ca2 site) has an eight-fold coordination [15]. When activated by Eu^3+^ ions, the ions enter both Ca1 and Ca2 sites to generate ^5^D_0_–^7^F_0_ emission peaks at 270 nm excitation [16].

In addition to the five types of polymorphs previously mentioned, an incommensurate (IC) phase has been reported to occur for the P_2_O_5_-doped C_2_S [17,18,19,20]. The modulation wavevector, *N*^−1^ × **b***, has been along the *b*-axis of the underlying orthorhombic basic structure, where *N* is a noninteger. Saalfeld and Klaska prepared the IC-phase with *N* = 3.75, and determined the atom arrangements within the hypothetical supercell of 4 × *b* based on the space group *Pnm*2_1_ [17]. Recently, the incommensurately modulated crystal structure of (Ca_1.88_Eu^2+^_0.01_☐_0.11_)(Si_0.78_P_0.22_)O_4_ (*N* = 3.649) [20], where ☐ denotes a vacancy in Ca site, has been determined, using a (3 + 1)-dimensional description based on the superspace formalism [21,22]. 

In the present study we have, for the first time, clarified the PL properties of the Eu^3+^-activated IC-phase phosphor, the chemical formula of which was (Ca_1.950_Eu^3+^_0.013_☐_0.037_)(Si_0.940_P_0.060_)O_4_. The incommensurately modulated crystal structure was determined from the X-ray powder diffraction data using a (3 + 1)-dimensional description based on the superspace group *Pnma*(0*β*0)00*s*. The sintered specimen was subject to different heat treatments in air at 1473–1773 K. We demonstrated a close relationship between the phase composition (consisting of β, α’_L_, and IC) and PL intensity.

## 2. Materials and Methods 

The materials were synthesized by solid-state reaction. A powder mixture with atom ratios of [Ca:Eu:Si:P] = [1.950:0.013:0.940:0.060] was prepared from reagent chemicals of CaCO_3_, SiO_2_, CaHPO_4_·2H_2_O, and Eu_2_O_3_ (>99.9% grade). Together with a small amount of acetone, it was thoroughly mixed in a planetary ball mill (Pulverisette P-6, Fritsch, Germany). The ZrO_2_ balls, with 2 mm diameter, were used in the mill at a rotation speed of 400 rpm. The mixed specimens were pressed into pellets and heated at 498 K for 6 h, 973 K for 2 h, and then 1273 K for 8 h in air. The disc-shaped sintered specimens, the chemical formula of which corresponds to (Ca_1.950_Eu^3+^_0.013_☐_0.037_)(Si_0.940_P_0.060_)O_4_, were subsequently annealed at four different temperatures (1473, 1573, 1673, and 1773 K) for 4 h in air, followed by quenching in water. The stable phases during the annealing processes were α at 1773 K and α’_H_ at 1473–1673 K. We examined the specimens with a scanning electron microscope equipped with an energy-dispersive spectrometer to measure grain sizes and to confirm homogeneity.

Phase identification was made based on the X-ray diffraction (XRD) data (CuKα_1+2_), which were obtained using an X-ray powder diffractometer (RINT 2500, Rigaku Co., Ltd., Japan) operated at 40 kV and 200 mA. The phase compositions, as well as the crystal structure of the IC-phase, were determined by the Rietveld method [23] from the X-ray profile intensity data (CuKα_1_) collected at 298 K on another diffractometer (X’Pert PRO Alpha-1, PANalytical B.V., Almelo, the Netherlands), operated at 45 kV and 40 mA. We quantitatively determined the phase compositions of the samples using the phase-analysis method (2*θ* range of 25.0°–43.0°) based on Brindley’s procedure [24], the subroutine of which was implemented in the computer program RIETAN-FP [25] based on the structural parameters reported by Jost et al. [15] for the β-phase, Udagawa et al. [26] for the α’_L_-phase, and Saalfeld and Klaska [17] for the IC-phase having a supercell of 4 × *b*. We used the computer programs JANA2006 [27] for the detailed structural analysis of the IC-phase (2*θ* range of 5.010°–147.981°), and VESTA [28] for the crystal structure drawing. Distortion parameters for the coordination polyhedra of the β-phase were determined using the computer program IVTON [29].

Excitation and emission spectra were obtained using a fluorescence spectrophotometer (F-7000, HITACHI, Japan).

## 3. Results and Discussion

### 3.1. Constituent Phases and Crystal Structure of IC-Phase 

Figure 1 shows a series of XRD patterns of (Ca_1.950_Eu^3+^_0.013_☐_0.037_)(Si_0.940_P_0.060_)O_4_ phosphors annealed at different temperatures. For the specimen annealed at 1473 K, the phase composition was 70.7 mol % β and 29.3 mol % α’_L_ (Figure 2a). The specimen annealed at 1773 K was composed exclusively of the IC-phase, as evidenced from the satisfactory pattern fitting result (Figure 2b), although the structural model used was of the commensurate structure with the supercell of 4 × *b*. This implies that the modulation wavevector of the IC-phase should be close to 0.25 (=1/4) × **b***. In fact, we found it to be 0.2433(2) × **b*** as discussed later, and hence there were slight positional shifts in Figure 2b from the proper positions for the weak reflections (e.g., 330 and $\bar{3}\bar{3}$0 reflections with 2*θ* ≈ 31.05°, and 530 and $\bar{5}\bar{3}$0 reflections with 2*θ* ≈ 35.23°) ascribed to the superstructure. Based on the present structural model, the specimens annealed at 1573–1773 K were free from the α’_L_-phase. The relative amount of the IC-phase with respect to the β-phase steadily increased when the annealing temperature was increased from 1473 to 1773 K (Table 1). 

We refined the incommensurately modulated structure starting from the initial structural model (superspace group *Pnma*(0*β*0)00*s*) that was equivalent to the crystal structure of (Ca_1.88_Eu^2+^_0.01_☐_0.11_)(Si_0.78_P_0.22_)O_4_ [20]. The refinement resulted in the satisfactory reliability indices of *R*(all) = 0.0326, *Rw*(all) = 0.0413, and *S*(all) = 2.99 (Appendix A). The crystallographic data and the structural parameters are summarized in Table 2 and Table 3, respectively. The basic structure contains two nonequivalent Ca sites, Ca1 and Ca2. In the average structure, the site occupation factors of both sites were fixed at 0.9750 for Ca, 0.0065 for Eu, and 0.0185 for vacancy. We tried to demonstrate the occupational modulation in all ranges of *t* (= *x*_4_ – **q** × **r**, where *x*_4_ is the fractional coordinate of the 4th direction in superspace description, **q** is the modulation wavevector, and **r** is the positional vector) at these sites to clarify the locations where the Eu^3+^ ions concentrate. However, it remained unclear, because the bond valence sums are nearly the same between Ca^2+^ and Eu^3+^. Figure 3 shows a partial structure of IC-(Ca_1.950_Eu^3+^_0.013_☐_0.037_)(Si_0.940_P_0.060_)O_4_. The *M* site, which is occupied by Si and P, is coordinated by 4 O ions to form an isolated [*M*O_4_] tetrahedron. The ratio of P/(P + Si) is fixed at 0.06 for the *M* site.

The present crystal structure is in accord with that of IC-(Ca_1.88_Eu^2+^_0.01_☐_0.11_)(Si_0.78_P_0.22_)O_4_, characterized by the tilting of [*M*O_4_] tetrahedra [20], and hence it can be regarded as being made up of five types of layers (denoted by *S*, *S*’’, *T*, *T*’’, and *U*) that include the [*M*O_4_] tetrahedra with different tilting directions and angles (= Θ) (Appendix A). With layer *S*, the tilting directions of the constituent [*M*O_4_] tetrahedra are, when viewed along [$\bar{1}$00], clockwise with 5° ≤ Θ < 16°. The other type of layer, *T*, contains tetrahedra of counterclockwise tilting and −16° < Θ ≤ −5°. The slightly tilted [*M*O_4_] tetrahedra, with |Θ| < 5°, is characteristic of the interlayer *U*. The atom arrangements of layer *S* and those of *T*, which are similar to the partial crystal structure of β-Ca_2_SiO_4_ [15], are approximately related by a mirror plane perpendicular to the *b*-axis of *Pnma*(0*β*0)00*s*. The thicknesses of these layers correspond to one-and-a-half the *d*_100_-value of β-Ca_2_SiO_4_. Although it is very rare, there are *S*’’ and *T*’’ layers with a thickness almost twice the *d*_100_-value. A repetitive sequence of the bundled *SUTU* layer eventually constructs the commensurate structure having the supercell of 4 × *b*. The present incommensurately modulated structure with the wavelength of 4.110 × *b* is formed by the occasional replacements of *S* with *S*’’ and/or *T* with *T*’’ in the commensurate structure (Appendix A). Since the modulated structures are definitely ordered ones, the connecting sequence of all these layers is strictly defined by the modulation function. In the crystal structure of IC-(Ca_1.88_Eu^2+^_0.01_☐_0.11_)(Si_0.78_P_0.22_)O_4_ with the modulation wavelength of 3.649 × *b* [20], there are two other layers of *S*’ and *T*’, the thicknesses of which are almost equal to the *d*_100_-value of β-Ca_2_SiO_4_. These layers have never been recognized in the present IC-phase. 

In previous studies, a negative linear correlation has been reported between the *N*- and P/(P+Si)-values for the P_2_O_5_-doped C_2_S [14,19]. The larger *N*-value for (Ca_1.950_Eu^3+^_0.013_☐_0.037_)(Si_0.940_P_0.060_)O_4_ (*N* = 4.110) than for (Ca_1.88_Eu^2+^_0.01_☐_0.11_)(Si_0.78_P_0.22_)O_4_ (*N* = 3.649) would be principally induced by the smaller P/(P+Si)-value of the former material as compared with the latter. 

### 3.2. PL Properties Associated with Phase Compositions

Figure 4 shows the emission and excitation spectra of the four types of Eu^3+^-activated phosphors annealed at 1473–1773 K. Under excitation at 394 nm, the red light was emitted due to the ^7^F_0_–^5^L_6_ intraconfigurational transition of Eu^3+^. The sharp and strong emission peaks appeared at around 594 nm (^5^D_0_–^7^F_1_, magnetic dipole), 625 nm (^5^D_0_–^7^F_2_, electric dipole), and 706 nm (^5^D_0_–^7^F_4_, electric dipole). 

In this paper, we controlled the phase compositions by annealing at different temperatures in order to investigate the relationship between the crystal structures and PL intensities. The PL intensity steadily decreased with increasing annealing temperature (Figure 4). This implied that the phase compositions of the Eu^3+^-activated phosphors readily affected the PL intensities. Actually, the intensity was highest for the specimen annealed at 1473 K, the constituent phases of which were β and α’_L_, as described previously. With increasing relative amount of the IC-phase with respect to the coexisting β-phase (Table 1), the PL intensity of the phosphor steadily decreased to reach the minimum, at which the specimen was composed exclusively of the IC-phase. As reported previously [8], the PL intensity has also been related to the relative amount of the α’_L_ phase with respect to the β phase.

In previous studies for the Eu^3+^-activated lithium tantalite-based phosphors, the centroid-to-cation distance (=eccentricity) of [(Li, Eu)O_12_] polyhedra was associated with increased efficiency of the red light emission [9,10]. The larger the magnitude of eccentricity became, the stronger the PL intensity of the emission became. In a similar manner, the eccentricity of the Eu^3+^ position in the crystal structure of P_2_O_5_-doped C_2_S phosphors could also be closely related to the PL intensity. We hypothesized that the eccentricity values of [(Ca, Eu)O*_n_*] polyhedra (*n* is the coordination number) in IC-(Ca_1.950_Eu^3+^_0.013_☐_0.037_)(Si_0.940_P_0.060_)O_4_ might be smaller compared to those of the Eu^3+^-activated β-phase, which would eventually cause a much weaker PL intensity for the former than for the latter. 

With β-Ca_2_SiO_4_, the Eu^3+^ ions occupied both Ca1 and Ca2 sites [15]. The eccentricity values were 0.044 and 0.022 nm for the polyhedra [Ca1O_7_] and [Ca2O_8_], respectively. The maximum bond lengths were 0.280 nm for Ca1–O and 0.279 nm for Ca2–O. With the present IC-phase, the Ca1 and Ca2 sites in the layer *U* were located at *t* = 0.189 and 0.689. They formed [Ca1O*_n_*] and [Ca2O*_n_*] polyhedra with *n* = 6, 7, or 8. The Ca sites, situated in the middle parts of the layers *S* (*t* = 0.439) or *T* (*t* = 0.939), formed the polyhedra of [Ca1O*_n_*] (*n* = 7) and [Ca2O*_n_*] (*n* = 8). The eccentricity values of these polyhedra with different *n*- and *t*-values are summarized in Table 4. With Ca1 sites, the eccentricity values were larger for the β-phase (=0.044 nm) than for the IC-phase at *t* = 0.689 (=0.031 nm). Furthermore, the eccentricity value of the IC-phase at *t* = 0.189 (=0.009 nm) was much smaller than that of the β-phase (=0.022 nm) for the Ca2 sites. Accordingly, the Eu^3+^ ions that contribute to the relatively weak red light emission intensity of the IC-phase could preferentially occupy those Ca sites. 

## 4. Conclusions

Four types of red-light-emitting phosphors, with different phase compositions but identical chemical composition, were prepared. A close relationship was suggested between the coordination environments of the Eu^3+^ ion in the crystal structures of the β- and IC-phases and their PL properties.
The β- and α’_L_-phases coexisted for the specimen annealed at 1473 K. With increasing annealing temperature, the relative amount of the IC-phase with respect to the β-phase steadily increased. The specimen annealed at 1773 K was composed exclusively of the IC-phase.The incommensurately modulated crystal structure was determined using a (3 + 1)-dimensional description based on the superspace group *Pnma*(0*β*0)00*s*. It was composed of β-phase like layers (*S*, *S*’’, *T*, and *T*’’) and the interlayer (*U*). The *S* and *S*’’ layers included the [*M*O_4_] tetrahedra (*M* = Si or P) tilted clockwise when viewed along [$\bar{1}$00]. Those of *T* and *T*’’ contained the tetrahedra tilted counterclockwise. The interlayer *U* was characterized by a slightly tilted tetrahedra. The incommensurate modulation, with modulation wavevector of 0.2433(2) × **b***, was induced by the long-range stacking order of these layers.The PL intensity was the highest for the specimen consisting of both β- and α’_L_-phases. As the relative amount of the IC-phase increased with respect to the coexisting β-phase, the PL intensity steadily decreased. The eccentricity of the Eu^3+^ position in the crystal structures of the β- and IC-phases could be closely related to the PL intensities. 

## Figures and Tables

**Figure 1 materials-13-00058-f001:**
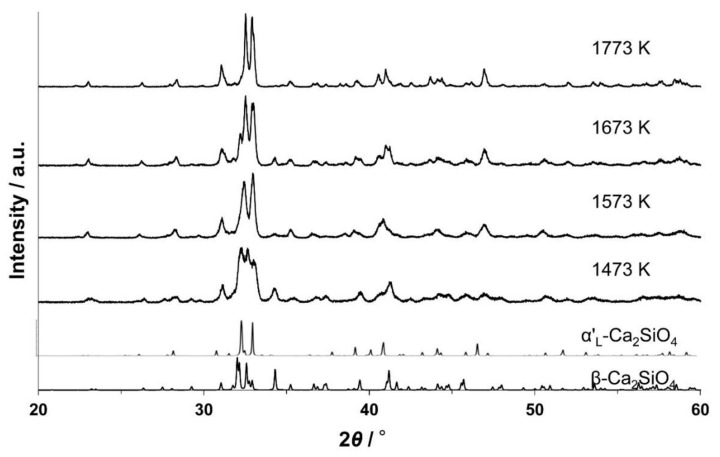
X-ray diffraction (XRD) patterns of samples annealed at 1473–1773 K.

**Figure 2 materials-13-00058-f002:**
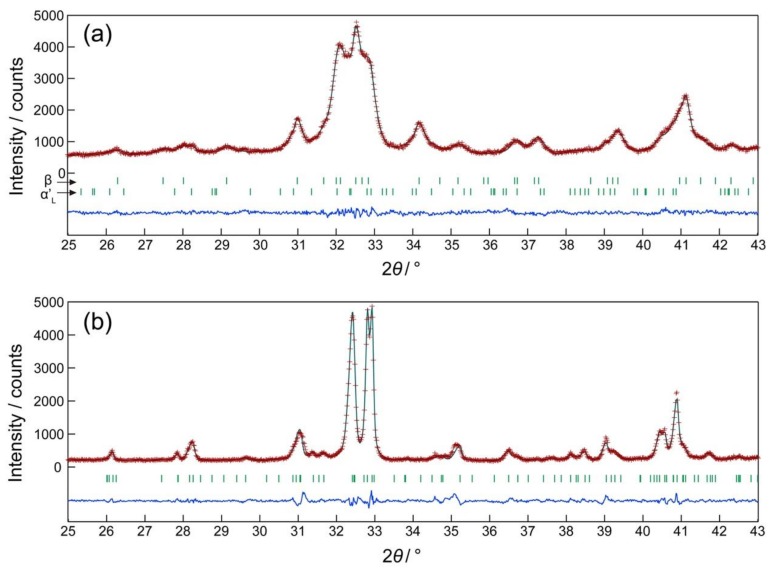
Comparison of the observed diffraction patterns (symbol: +) with the corresponding calculated patterns (upper solid line). The difference curve is shown in the lower part of each diagram. The vertical bars indicate the positions of possible Bragg reflections. The profile intensities were collected for samples annealed at (**a**) 1473 K and (**b**) 1773 K.

**Figure 3 materials-13-00058-f003:**
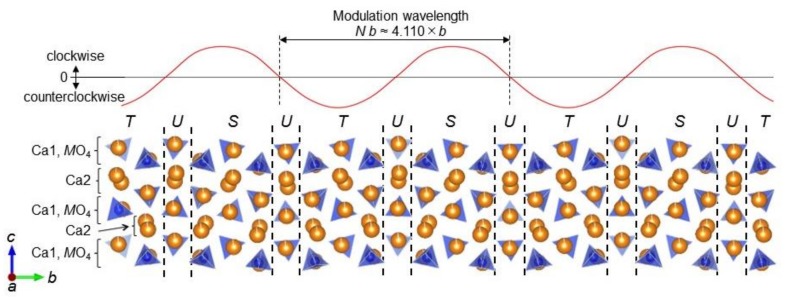
Projection of a partial structure along the *a*-axis.

**Figure 4 materials-13-00058-f004:**
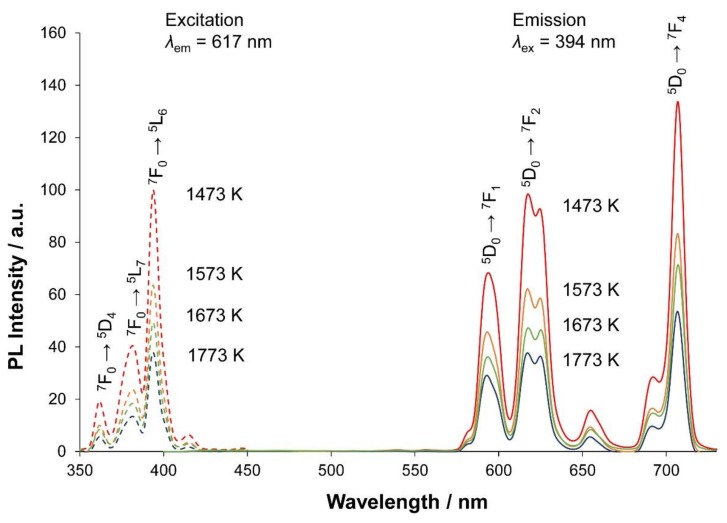
Emission and excitation spectra of the samples annealed at 1473–1773 K.

**Table 1 materials-13-00058-t001:** Changes in phase compositions for the samples annealed at 1473–1773 K. IC: incommensurate.

	Phase	β	α’_L_	IC
K				
1473		70.7 mol %	29.3 mol %	-
1573		18.9 mol %	-	81.1 mol %
1673		8.5 mol %	-	91.5 mol %
1773		-	-	100 mol %

**Table 2 materials-13-00058-t002:** Crystallographic data of IC-phase.

Chemical Formula	(Ca_1.950_Eu^3+^_0.013_☐_0.037_)(Si_0.940_P_0.060_)O_4_
Crystal system	Orthorhombic
Superspace group	*Pnma*(0*β*0)00*s*
*a*/nm	0.68004(2)
*b*/nm	0.54481(2)
*c*/nm	0.93956(3)
Modulation wavevector	0.2433(2) × **b***
*V*/nm^3^	0.34810(2)
*Z*	4
*D_x_*/Mgm^−3^	3.289

**Table 3 materials-13-00058-t003:** Structural parameters of the basic structure for IC-(Ca_1.950_Eu^3+^_0.013_☐_0.037_)(Si_0.940_P_0.060_)O_4._

Site	*x*	*y*	*z*	*U*_iso_ (×10^−2^ nm^2^)	sof
Ca1	0.1680(3)	1/4	0.4243(4)	0.0270(10)	Ca/Eu: 0.9750/0.0065
Ca2	0.4935(3)	1/4	0.7106(2)	0.0139(7)	Ca/Eu: 0.9750/0.0065
*M*	0.2240(4)	1/4	0.0755(6)	0.0134(10)	Si/P: 0.940/0.060
O1	0.3213(12)	1/4	0.9305(9)	0.025(3)	1
O2	0.2994(8)	0.0092(12)	0.1471(7)	0.0181(18)	1
O3	−0.0059(10)	1/4	0.0731(13)	0.038(3)	1

**Table 4 materials-13-00058-t004:** Centroid-to-cation distance (eccentricity) of [CaO*_n_*] polyhedra for IC-(Ca_1.950_Eu^3+^_0.013_☐_0.037_)(Si_0.940_P_0.060_)O_4._

Site	*t*	*n*	Ca–O Max. (nm)	Eccentricity (nm)	Layer
Ca1	0.189	7	0.280	0.054	*U*
Ca1	0.189	6	0.267	0.105	*U*
Ca1	0.689	8	0.294	0.041	*U*
Ca1	0.689	7	0.273	0.031	*U*
Ca1	0.439	7	0.280	0.041	*S*
Ca1	0.939	7	0.280	0.041	*T*
Ca2	0.189	7	0.259	0.009	*U*
Ca2	0.689	6	0.258	0.040	*U*
Ca2	0.439	8	0.279	0.023	*S*
Ca2	0.939	8	0.279	0.023	*T*

Phase parameter *t* is defined by *t* = *x*_4_ − **q** × **r**, where *x*_4_ is the fractional coordinate of the 4th direction in (3 + 1)-dimensional superspace description, **q** is the modulation wavevector, and **r** is the positional vector.

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
