# Peer review of "Incommensurately Modulated Crystal Structure and Photoluminescence Properties of Eu2O3- and P2O5-Doped Ca2SiO4 Phosphor"

_materials, 2019, doi:10.3390/ma13010058_

Round 1

Reviewer 1 Report

Having examined your manuscript entitled “Incommensurately Modulated Crystal Structure and Photoluminescence Properties of Eu2O3- and P2O5-Doped Ca2SiO4 Phosphor

I note that you have made some interesting measurements related to the establishment of the incommensurate phase and their influence in the PL response but before their publication several issues should be addressed. Therefore, it requires a mayor revision. Please check carefully.

Abstract, introduction and Conclusions

The author should highlight the relevance to develop the incommensurate (IC) phase based materials and provide quantitative values regarding other calcium aluminate compounds to judge the values presented.

The authors should underline the innovative process, what is the advantage of the synthesis selected. Please give examples and compare with the state of the art.

Unfortunately, the materials and their properties are not compared with existing state-of-the-art compounds or standards making it difficult to judge what improvements they offer. Specifically, all the photoluminescence studies present data in arbitrary units and there is no quantitative comparison with the luminescence properties of commercial, or other synthesized materials. The reader therefore cannot adequately judge the merits of your synthetic process. Therefore your article will not be considered, if you don’t include a reference phosphor powder.

Lakshmi Devi, C.K. Jayasankar,Spectroscopic investigations on high efficiency deep red-emitting Ca2SiO4:Eu3+ phosphors synthesized from agricultural waste,Ceramics International,Volume 44, Issue 12,2018,Pages 14063-14069, https://doi.org/10.1016/j.ceramint.2018.05.003.

2. Nakano, K. Kamimoto, N. Yokoyama, K. Fukuda, The effect of heat treatment on the emission color of p-doped Ca2SiO4 phosphor, Materials (Basel). 10 (2017). doi:10.3390/ma10091000.

Previously, a comparative study has been done by other authors as:

3. R.E. Rojas-Hernandez, F. Rubio-Marcos, A. Serrano, E. Salas, I. Hussainova, J.F. Fernandez, Towards blue long-lasting luminescence of Eu/Nd-doped calcium-aluminate nanostructured platelets via the molten salt route, Nanomaterials. 9 (2019). doi:10.3390/nano9101473.

The authors discuss vaguely the applicability of this kind of compounds, and the requirements particle size, morphology, stability needed.

The authors said that all compounds have the same chemical composition. This statement is not totally true, in principle you do the synthesis with the calculated amounts of each precursors, however after the synthesis maybe you don’t have the same chemical composition in all compound thermal treated at different temperatures. You have only assign the phase composition, but you don’t give anothers results that prove the same chemical composition.

Materials and Methods

What is the advantage and novelty of the synthesis process employed?

What is the particle size of the precursors selected

Please, include the rpm, ball sizes, relation between medium, powder, balls  of planetary ball mill process done by  Pulverisette P-6,  Fritsch, Germany

Results and Discussion section:

In Figure 1. Please include all the phase assignments for all the peaks showed. And add the corresponding Powder Diffraction File (number and reflections).

In the caption of Fig 2. You said that there are + symbols, but you did not include it.

The author so a correlation between the PL response with the presence of the IC-phase. However, the author don’t give any information related how the crystallinity, grain size particle size have influence in the PL properties.

The best perfiormance is obtained when the The beta- and alpha’L-phases coesxit. So which phase is the best one? There is any information reported about the response of each phase.

In section 3.2. The authors showed the excitation and emission spectra. Please assign each peak presented and the broad band emission in the Figures. Why the author selected the excitation wavelength 394nm.

The author focus the paper on the IC phase that shows the lowest PL response. Therefore, what is the advantage to obtain this phase?

You should include SEM micrographs to verify the homogeneity of the powder. etc

Therefore your article will not be considered, if the author does not include and add the changes proposed.

Author Response

Having examined your manuscript entitled “Incommensurately Modulated Crystal Structure and Photoluminescence Properties of Eu2O3- and P2O5-Doped Ca2SiO4 Phosphor

I note that you have made some interesting measurements related to the establishment of the incommensurate phase and their influence in the PL response but before their publication several issues should be addressed. Therefore, it requires a mayor revision. Please check carefully.

Thank you for your comments. We have done our best to answer all comments, and revised the paper accordingly. We have added new sentences using blue ink in the paper.

Abstract, introduction and Conclusions

The author should highlight the relevance to develop the incommensurate (IC) phase based materials and provide quantitative values regarding other calcium aluminate compounds to judge the values presented.

 Answer: Calcium aluminate compounds [A] possess different crystal structures. Therefore, we will not discuss the features of IC phase in light of those of calcium aluminate compounds in the present paper.

[A] Acta Cryst. (2010). B66, 585-593 https://doi.org/10.1107/S0108768110035792

The authors should underline the innovative process, what is the advantage of the synthesis selected. Please give examples and compare with the state of the art.

Answer: This paper is not concerned with an innovative process, and the materials have been synthesized by normal solid-state reaction. We have focus on the relationship between the PL intensities and crystal structures of calcium silicate phosphors. We have added a new sentence in Section 2 using blue ink as follows:

Sentence: The materials were synthesized by solid-state reaction.

Unfortunatelythe materials and their properties are not compared with existing state-of-the-art compounds or standards making it difficult to judge what improvements they offer. Specifically, all the photoluminescence studies present data in arbitrary units and there is no quantitative comparison with the luminescence properties of commercial, or other synthesized materials. The reader therefore cannot adequately judge the merits of your synthetic process. Therefore your article will not be considered, if you don’t include a reference phosphor powder.

1.Lakshmi Devi, C.K. Jayasankar,Spectroscopic investigations on high efficiency deep red-emitting Ca2SiO4:Eu3+ phosphors synthesized from agricultural waste,Ceramics International,Volume 44, Issue 12,2018,Pages 14063-14069, https://doi.org/10.1016/j.ceramint.2018.05.003.

H. Nakano, K. Kamimoto, N. Yokoyama, K. Fukuda, The effect of heat treatment on the emission color of p-doped Ca2SiO4 phosphor, Materials (Basel). 10 (2017). doi:10.3390/ma10091000.

Previously, a comparative study has been done by other authors as:

R.E. Rojas-Hernandez, F. Rubio-Marcos, A. Serrano, E. Salas, I. Hussainova, J.F. Fernandez, Towards blue long-lasting luminescence of Eu/Nd-doped calcium-aluminate nanostructured platelets via the molten salt route, Nanomaterials. 9 (2019). doi:10.3390/nano9101473.

Answer: We think that the photoluminescence data in arbitrary units is standard style.              As the quenching process is well known as one of the annealing processes for ceramics, we did not discuss the processing route nor its merit in this paper. In this paper, we wanted to stress the relationship between PL intensity and crystal structure. We believe that readers will understand the relationship and can apply it for the design of new phosphors. The reference (No.2) is our previous paper and it was listed in the paper as No. 8.

The authors discuss vaguely the applicability of this kind of compounds, and the requirements particle size, morphology, stability needed.

Answer: If specimens have the same crystal structures of host materials, PL intensities are mainly affected by particle size and morphologies. We observed the grain sizes from 4 to 7 µm after the annealing treatments. However, in this paper, we controlled the phase compositions by annealing to discuss the relationship between crystal structures and PL intensities. To clarify this, we have added a new sentence in section 2 and 3.2, respectively using blue ink.

Sentence:

In Section 2:

We examined the specimens by scanning electron microscope equipped with an energy-dispersive spectrometer to measure grain sizes and confirm homogeneity.

In Section 3.2:

We controlled the phase compositions by annealing at different temperatures in order to discuss the relationship between the crystal structures and PL intensities.

The authors said that all compounds have the same chemical composition. This statement is not totally true, in principle you do the synthesis with the calculated amounts of each precursors, however after the synthesis maybe you don’t have the same chemical composition in all compound thermal treated at different temperatures. You have only assign the phase composition, but you don’t give anothers results that prove the same chemical composition.

Answer: The chemical formula was determined from calculations based on the starting materials. We synthesized the samples using a normal solid-state reaction method, without any precursors.

Materials and Methods

What is the advantage and novelty of the synthesis process employed?

Answer: We used the classical method, which enabled us to easily control the crystal structures of the resulting materials. Actually, the crystal structures of the calcium silicate phases have changed after the annealing process.

What is the particle size of the precursors selected

Answer: As mentioned above, in this paper, the normal solid-state reaction method was used. Therefore, we did not use any precursors.

Please, include the rpm, ball sizes, relation between medium, powder, balls of planetary ball mill process done by Pulverisette P-6,  Fritsch, Germany

Answer: We have added two new sentences in Section 2 as follows:

Sentences: The ZrO2 balls, with 2 mm diameter, were used in the mill, at a rotation speed of 400 rpm. The mixed specimens were pressed into pellets,

Results and Discussion section:

In Figure 1. Please include all the phase assignments for all the peaks showed. And add the corresponding Powder Diffraction File (number and reflections).

Answer: For the multiphase samples, the phase assignment is more complex in the figure. Therefore, we have shown the XRD profiles of the a’L and b phases from the data-base in a new Fig. 1. We have determined the phase compositions using a RIETAN program.

In the caption of Fig 2. You said that there are + symbols, but you did not include it.

Answer: We have plotted the peaks using larger + symbols in the revised Figure 2.

The author so a correlation between the PL response with the presence of the IC-phase. However, the author don’t give any information related how the crystallinity, grain size particle size have influence in the PL properties.

Answer: As described above, the PL intensities are affected by particle size and morphology for the same crystal structures of host materials. In this case, we have focused on the relationship between crystal structures and PL intensities, and concluded that the structural changes would be a more effective factor for the PL intensities. We added the sentence as follows in 3.2.

Sentence: We controlled the phase compositions by annealing at different temperatures in order to discuss the relationship between the crystal structures and PL intensities.

The best performance is obtained when the the beta- and alpha’L-phases coexist. So which phase is the best one? There is any information reported about the response of each phase.

Answer: We determined the relationship between phase compositions and PL intensities previously [8]. Hence, we have added a new sentence in 3.2 as follows:

Sentence: As reported previously [8], the PL intensity has also been related to the relative amount of the a’L phase with respect to the b phase.

In section 3.2. The authors showed the excitation and emission spectra. Please assign each peak presented and the broad band emission in the Figures. Why the author selected the excitation wavelength 394nm.

Answer: We have assigned each peak in a new Fig. 5. In industrial use, red-emission phosphors have been used at an excitation wavelength around 400 nm.

The authors focus the paper on the IC phase that shows the lowest PL response. Therefore, what is the advantage to obtain this phase?

Answer: We have not focused on the merit of the IC phase. As described before many times, we discussed the relationship between the PL intensities and crystal structures of calcium silicate phosphors. We discussed the unique crystal structural changes as well as the relationship between crystal structures and PL intensities. In general, complex phases such as the IC phase cannot be determined easily by XRD. However, we succeeded in the characterization of the IC phase through our collaboration with Dr. Michiue, who is one of the top scientists in crystallography.

You should include SEM micrographs to verify the homogeneity of the powder. Etc Therefore your article will not be considered, if the author does not include and add the changes proposed.

Answer: The SEM observation showing homogeneity is shown here. We focus on the relationship between the PL intensities and crystal structures of calcium silicate phosphors. We have added a new sentence in Section 2 as follows:

Sentence. We examined the specimens by scanning electron microscope equipped with an energy-dispersive spectrometer to measure grain size and confirm homogeneity.

Reviewer 2 Report

The article shows great thoroughness of structural studies of Ca2SiO4 compound. This is an advantage of this article but sometimes it can be also disadvantageous, while it sounds as if it was directed to very specialized readers in this particular narrow field of materials study. Some values appear in the text as if they are commonly known, but in fact there should be more explanation. Detailed comments are as following:

In section 1. – Introduction – Authors describe C2S polymorphs and transformation temperatures according to ref 11. I could not gain access to this book to check this values, but I know of another book in this subject - Encyclopedia of the Alkaline Earth Compounds by R.C. Ropp (Elsevier, 2013) https://doi.org/10.1016/B978-0-444-59550-8.00005-3 - and here are completely different values of phase change temperatures for Ca2SiO4 – 847°C (1120 K), 1300°C (1573 K) and 1437°C (1710 K). This changes the range where α’H phase is stable. According to this values sample annealed at 1473 K was in range of α’L phase, annealing at 1573 K is very close to α’L – α’H transition and sample annealed at 1673 K was in range of α’H phase. It explains why sample annealed at 1473 K has no trace of α’H phase and α’H phase occurs for samples annealed in 1573K and higher. I cannot say which temperatures are correct, and which are not (by H.F. Taylor form ref 11 or R.C. Ropp) however considering one of the Authors long experience in calcium silicates I would like to ask Authors for a comment to that problem.

In section 3. – Results and discussion – in line 119 of the paper Authors use the term “average structure”. What is meant by this term?

In section 3. – Results and discussion – in line 120-121 Authors write: “We tried to demonstrate the occupational modulation in all range of t (= x4 - q·r) at these sites…” however neither t nor x4 nor q nor r are explained in the text. It would help to understand t values in Table 3.

In section 3. – Results and discussion – in line 133 Authors mention five types of layers in the crystal structure, but label T” is not present on any figure in the text nor in supplementary materials. I believe that layer labelled as T in Figure S3 b) could be labelled as T”. Is it true?

In section 3. – Results and discussion – in line 150 Authors refer to P/(P+Si) value. I understand that it means the ratio of molar content of P to (P+Si). I think that it should be written plainly what Authors mean by a new value occurring in the text without the necessity to look for references. References are needed for readers who are more interested in the subject and would like to compare values and not for understanding the article itself.

In section 3. – Results and discussion – the first mention of Figure 3 occurs in line 165 – after mentioning figures 4 and 5. Why Authors chose such numeration of figures?

And I must admit that I am slightly bothered by this lines that connect the dots in Fig.3. This lines suggest linear character of changes of % phase composition with temperature but it is obviously not linear. Why not just leave the dots without connecting them or maybe present this results in the table or in other form of plot?

Author Response

The article shows great thoroughness of structural studies of Ca2SiO4 compound. This is an advantage of this article but sometimes it can be also disadvantageous, while it sounds as if it was directed to very specialized readers in this particular narrow field of materials study. Some values appear in the text as if they are commonly known, but in fact there should be more explanation. Detailed comments are as following:

Thank you for your comments. We have done our best to answer all comments, and revised the paper accordingly. We have added new sentences using blue ink in the paper.

In section 1. – Introduction – Authors describe C2S polymorphs and transformation temperatures according to ref 11. I could not gain access to this book to check this values, but I know of another book in this subject - Encyclopedia of the Alkaline Earth Compounds by R.C. Ropp (Elsevier, 2013) https://doi.org/10.1016/B978-0-444-59550-8.00005-3 - and here are completely different values of phase change temperatures for Ca2SiO4 – 847°C (1120 K), 1300°C (1573 K) and 1437°C (1710 K). This changes the range where α’H phase is stable. According to this values sample annealed at 1473 K was in range of α’L phase, annealing at 1573 K is very close to α’L – α’H transition and sample annealed at 1673 K was in range of α’Hphase. It explains why sample annealed at 1473 K has no trace of α’Hphase and α’H phase occurs for samples annealed in 1573K and higher. I cannot say which temperatures are correct, and which are not (by H.F. Taylor form ref 11 or R.C. Ropp) however considering one of the Authors long experience in calcium silicates I would like to ask Authors for a comment to that problem.

Answer: The corresponding literature (reference 11) says that the α’L-to-α’H temperature is 1160°C (1433 K) and the α’H-to-α temperature is 1425°C (1698 K), as mentioned in the manuscript. Thus, we think that the samples were annealed (at 1473 – 1773 K) in the stable temperature ranges of the α (1773 K) and α’H (1473 – 1673 K) phases.

Sentence: In Section 3.1

We tried to demonstrate the occupational modulation in all range of t (= x4q·r, where x4 is the fractional coordinate of the 4th direction in superspace description, q is the modulation wavevector, and r is the positional vector) at these sites to clarify the locations where

In section 3. – Results and discussion – in line 119 of the paper Authors use the term “average structure”. What is meant by this term?

Answer: The phrase “In the average structure” was deleted in the revised manuscript, because “the average structure” has almost the same meaning as “the basic structure” in the preceding sentence.

In section 3. – Results and discussion – in line 120-121 Authors write: “We tried to demonstrate the occupational modulation in all range of t (= x4 - q·r) at these sites…” however neither t nor x4 nor q nor r are explained in the text. It would help to understand t values in Table 3.

Answer: A definition of the phase parameter t, and explanations of x4, q, and r, have been given in the text, and also in the footnote to Table 4 and the caption for Figure S2 in the revised manuscript.

Footnote: Phase parameter t is defined by t = x4q·r, where x4 is the fractional coordinate of the 4th direction in (3+1) dimensional superspace description, q is the modulation wavevector, and r is the positional vector.

In section 3. – Results and discussion – in line 133 Authors mention five types of layers in the crystal structure, but label T” is not present on any figure in the text nor in supplementary materials. I believe that layer labelled as T in Figure S3 b) could be labelled as T”. Is it true?

Answer: No layer labelled as T in Figure S3 can be labelled as T”. Therefore, supplementary material Figure S3(c) has been added, where we give an example of the replacement of layer T with layer T”, which is not in (a) but elsewhere in the crystal.

Fig.S3 (c) An example of the replacement of layer T with layer T”.

In section 3. – Results and discussion – in line 150 Authors refer to P/(P+Si) value. I understand that it means the ratio of molar content of P to (P+Si). I think that it should be written plainly what Authors mean by a new value occurring in the text without the necessity to look for references. References are needed for readers who are more interested in the subject and would like to compare values and not for understanding the article itself.

 Answer: We added the P/(P+Si) value in the paper.

Sentence: The ratio of P/(P + Si) is fixed at 0.06 for the M site.

In section 3. – Results and discussion – the first mention of Figure 3 occurs in line 165 – after mentioning figures 4 and 5. Why Authors chose such numeration of figures?

Answer: We checked the Figure No. The first mention of Fig. 3 occurs in line 123 (original paper).

And I must admit that I am slightly bothered by this lines that connect the dots in Fig.3. This lines suggest linear character of changes of % phase composition with temperature but it is obviously not linear. Why not just leave the dots without connecting them or maybe present this results in the table or in other form of plot?

Answer: We have revised Table 1 as suggested, and in following Figure numbers and Table numbers.

Round 2

Reviewer 1 Report

The authors replied the questions and add some inputs proposed. 

Therefore, the article can be accepted. 

Reviewer 2 Report

I think that the revised version of the manuscript is better than the original paper, and it meets the criteria for publishing. I still think that some comment on the phase transition temperatures (different sources gives different phase transition temperatures) would be helpful and would explain better why some of the samples have traces of other phases. It would surely improve the scientific quality of the article, however it does not change the fact, that experiments were performed correctly and results are analysed properly.